# Direct structural observation of ultrafast photoisomerization dynamics in sinapate esters

Temitope T. Abiola [1], Josene M. Toldo [2✉], Mariana T. do Casal[2], Amandine L. Flourat [3], Benjamin Rioux [3], Jack M. Woolley [1], Daniel Murdock[1], Florent Allais [3✉], Mario Barbatti [2,4] & Vasilios G. Stavros [1✉]

Sinapate esters have been extensively studied for their potential application in 'nature-inspired' photoprotection. There is general consensus that the relaxation mechanism of sinapate esters following photoexcitation with ultraviolet radiation is mediated by geometric isomerization. This has been largely inferred through indirect studies involving transient electronic absorption spectroscopy in conjunction with steady-state spectroscopies. However, to-date, there is no direct experimental evidence tracking the formation of the photo-isomer in real-time. Using transient vibrational absorption spectroscopy, we report on the direct structural changes that occur upon photoexcitation, resulting in the photoisomer formation. Our mechanistic analysis predicts that, from the photoprepared $\pi\pi^\star$ state, internal conversion takes place through a conical intersection (CI) near the geometry of the initial isomer. Our calculations suggest that different CI topographies at relevant points on the seam of intersection may influence the isomerization yield. Altogether, we provide compelling evidence suggesting that a sinapate ester's geometric isomerization can be a more complex dynamical process than originally thought.

[1] Department of Chemistry, University of Warwick, Gibbet Hill Road, Coventry CV4 7AL, UK. [2] Aix Marseille Université, CNRS, ICR, Marseille, France. [3] URD Agro-Biotechnologies (ABI), CEBB, AgroParisTech, 51110 Pomacle, France. [4] Institut Universitaire de France, 75231 Paris, France. ✉email: josene-maria.toldo@univ-amu.fr; florent.allais@agroparistech.fr; v.stavros@warwick.ac.uk

Photoinduced isomerization reactions (e.g., *trans→cis*, denoted as *E→Z* henceforth) offer efficient relaxation mechanisms mediated via internal conversion (IC), yielding either the starting isomer or its photoisomer counterpart[1–4]. Due to this efficient relaxation mechanism, photoisomerization has become a targeted photoinduced reaction mechanism where efficient and safe dissipation of absorbed energy is required, playing vital role in the design and application of many photosystems, including photoswitches[5–8], photoreceptor proteins[9–17], phototherapy[18], molecular motors[19,20] and photoprotection[21–26]. Previous studies have reported that photoisomerization occurs on an ultrafast timescale, within femtoseconds (fs) to picoseconds (ps). For example, retinal in the photoreceptor protein rhodopsin with 11-*Z* configuration has been extensively studied and reported to adopt an all-*E* geometry within ~200 fs following photoexcitation[10,27]. The well-studied photoswitch azobenzene has also been reported to complete *E→Z* photoisomerization in solution within ~16 ps[28,29].

Another class of molecules for which photoisomerization has been reported as the main relaxation pathway is the sinapate esters (Fig. 1)[21,24,25,30–32]. These molecules (and derivatives thereof) have attracted considerable attention due to their broad spectral absorption in the ultraviolet (UV) region, high extinction coefficient, and potential application in sunscreen formulations and photothermal materials[33,34]. Their photophysics have been extensively studied to better understand the dynamics that ultimately lead to efficient dissipation of absorbed energy to the surroundings as heat. In short, photoexcitation of this class of molecules typically populates the lowest singlet excited-state ($S_1$, $1^1\pi\pi^*$) from where they evolve through a proposed *E→Z* isomerization channel across the allylic C=C double bond[21,24,25,30–32]. Early studies have invoked a stepwise mechanism of relaxation for sinapate esters involving initial $S_1$ population traversing to the $S_2$ ($2^1\pi\pi^*$) state; the molecule subsequently relaxes back to the ground-state via a conical intersection (CI) between the $S_2$ and the electronic ground-state ($S_0$), labelled $S_2/S_0$ CI[21]. Subsequent studies by Zhao et al. suggested, however, that the $S_2$ state is not involved in the relaxation of a series of sinapate esters. The authors instead proposed that *E→Z* photoisomerization is mediated by a $S_1/S_0$ CI, from which branching occurs to form both the starting isomer and the photoisomer in their respective $S_0$ states[35]. In contrast, a recent study suggested a competing nonadiabatic and adiabatic relaxation of the excited-state population along the potential energy surface of *Z*-methyl sinapate in methanol[32].

Despite the extensive experimental and theoretical efforts to elucidate the stepwise relaxation pathways resulting in photoisomerization dynamics, there is still an ongoing debate on the relaxation pathways of sinapate esters, as well as on the interpretation of many other aspects of their excited-state dynamics[21,25,31,35]. This debate includes whether photoisomerization occurs in the excited-state or in the ground-state following IC; this uncertainty is a consequence of inferring the isomerization pathways indirectly, since direct structural changes are not observed in these measurements.

In the present work, we utilize transient vibrational absorption spectroscopy (TVAS) to directly probe the excited-state vibrational levels of the separate *E* and *Z* isomers of ethyl sinapate (ES, Fig. 1) in solution, and to measure the evolution of vibrational structure over time. Our results are complemented with computational studies, providing strong evidence that photoisomerization occurs in the ground-state after population traverses through the CI. Importantly, we find that IC occurs at multiple points along the potential energy surface: there are two CIs, one closer to the Franck-Condon region with a smaller torsional angle, and another, further away from the Franck-Condon region with a larger torsional angle, both of which facilitate the *E→Z* isomerization that results in the formation of the photoisomer.

## Results

**Steady-state infrared spectroscopy**. The FTIR spectra obtained in acetonitrile for both *E*-ES (Fig. 1a) and *Z*-ES (Fig. 1b) before and after irradiation are displayed in Fig. 1. Additional FTIR data are shown in the supplementary information Fig. S1 and discussed in Note S1. Frequency calculations (discussed in the supplementary information, Note S2 and Table S1) reveal that the vibrational mode centered at ~1710 cm$^{-1}$ in both isomers corresponds to a stretching motion of the carbonyl (C=O) group. We focus on this vibrational mode for the remainder of this work due to its transition intensity. For *E*-ES, the C=O stretch appears at 1707 cm$^{-1}$ before irradiation with a solar simulator and is spectrally blue-shifted to 1710 cm$^{-1}$, with reduced signal intensity following irradiation (see Fig. 1). Contrarily, for *Z*-ES, the C=O stretch appears at 1713 cm$^{-1}$ before irradiation and is spectrally red-shifted to 1710 cm$^{-1}$, with increased intensity following irradiation. This suggests that, after irradiation of either starting isomer, both *E*-ES and *Z*-ES are present in the solution, which accounts for the shift in wavenumber and changes in the intensity of the FTIR spectra. Additional FTIR spectra obtained in deuterated chloroform, demonstrating the effect of solvent

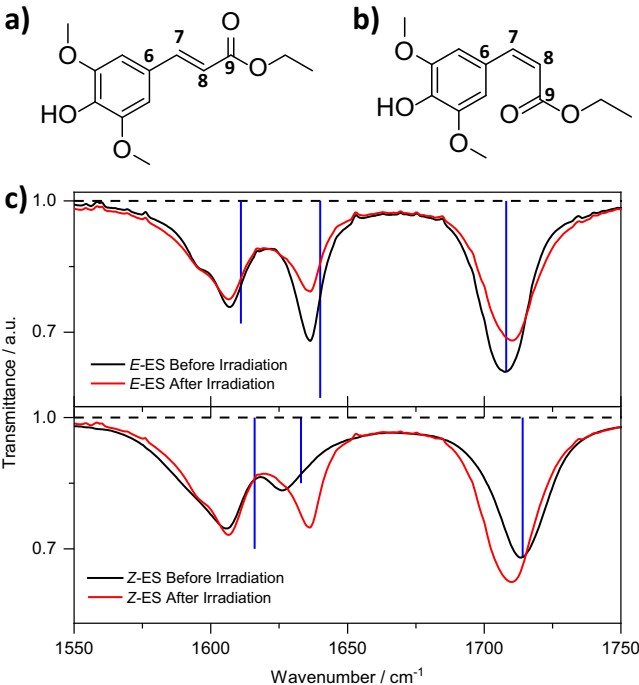

**Fig. 1 Structure and vibrational spectra of ethyl sinapate (ES) isomers. a** *E*-ES molecular structure, and **b** *Z*-ES molecular structure. **c** Steady-state vibrational spectra of 20 mM solution of *E*-ES (top panel) and *Z*-ES (bottom panel) in acetonitrile before (black) and after (red) 6 h irradiation with a solar simulator with irradiance equivalent to 1 sun (1000 W/m²). The overlaid vertical blue lines in **c** correspond to the calculated frequencies at B3LYP/6-311++G** level of theory in implicit acetonitrile; the lines' wavenumbers have been scaled with factors of 0.9879 for *E*-ES and 0.9908 for *Z*-ES as further discussed in the supplementary information Note S2. Frequency calculations suggest that the vibrational modes centered at ~1605 cm$^{-1}$, ~1635 cm$^{-1}$, and ~1710 cm$^{-1}$ correspond to aromatic C−H bend and C=C stretch, allylic C=C stretch, and C=O stretch, respectively (see supplementary information Note S2 for details).

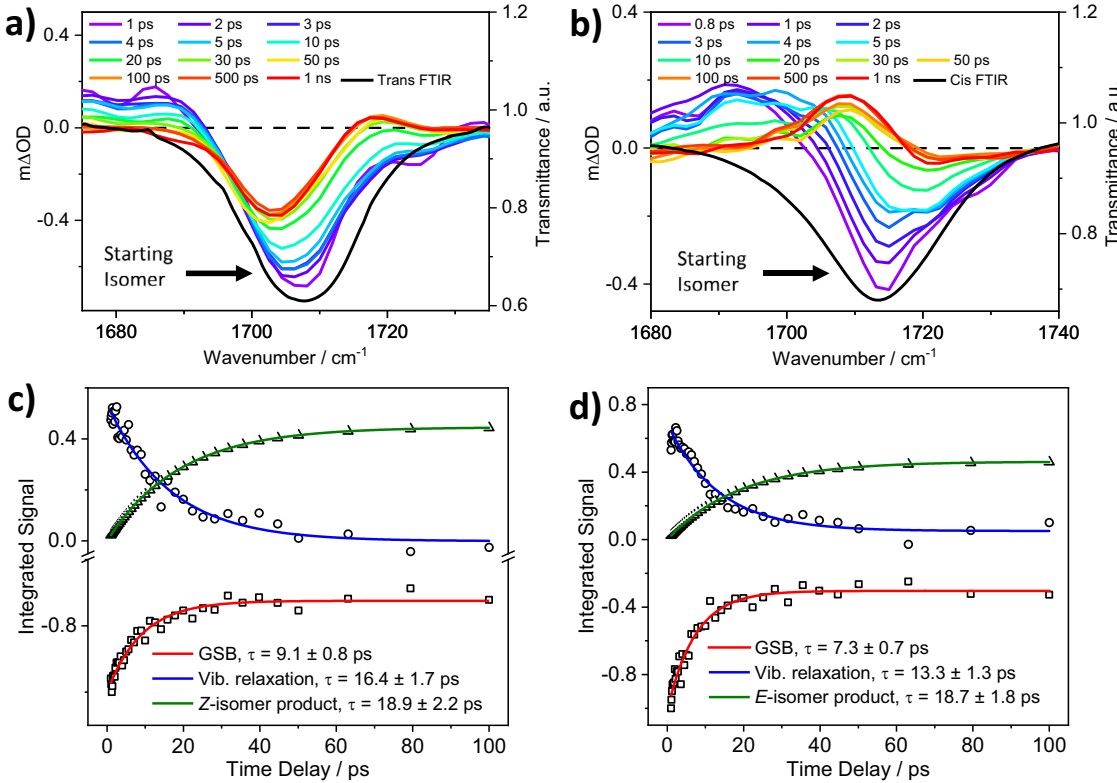

**Fig. 2 Transient vibrational absorption (TVA) spectra and exponential fittings of *E*-ES and *Z*-ES. a** *E*-ES after photoexcitation with 321 nm, and **b** *Z*-ES after photoexcitation with 318 nm, both in acetonitrile. The color key in panels **a** and **b** shows the different pump-probe time delays. For both TVA spectra, the probe pulse is centered on 1710 cm$^{-1}$. Kinetics of the ground-state bleach (GSB) recovery band of *E*-ES (*Z*-ES), centered at ~1707 (~1713) cm$^{-1}$, vibrational relaxation at ~1680 (~1690) cm$^{-1}$, and photoisomers at 1718 (1708) cm$^{-1}$, obtained using the KOALA software package are shown in **c** for *E*-ES and **d** for *Z*-ES.

environment in these dynamics, are shown in supplementary information Fig. S1.

**Transient vibrational absorption spectroscopy and dynamics.** Structural changes initiated by photoexcitation of IR-active vibrational modes are reflected on changes in the post-irradiation vibrational spectrum of the chromophore. A negative signal in TVAS data, termed a ground-state bleach (GSB), indicates loss of population in a specific IR-active mode upon photoexcitation. Conversely, positive signals may arise from a number of sources, including: (*i*) new vibrations of photoexcited molecules in the excited-state, (*ii*) vibrationally 'hot' $S_0$ molecules formed following IC from the excited-state, and/or (*iii*) vibration of photoproducts or intermediate species. Hence, photoproducts formed following photoexcitation of a molecule with IR-active vibrational modes can be identified in real-time by TVAS by the appearance and growth of positive features. As reported in previous TVAS data, photoproducts often appear as an excited-state absorption (ESA) at different wavenumbers (compared to the GSB band) and an incomplete GSB recovery of the original molecule's vibrational mode[36]. We observed this typical behavior in our solution-phase measurements of separate samples of *E*-ES or *Z*-ES using TVAS, and therefore conclude that photoinduced isomerization takes place.

Figure 2 shows transient vibrational absorption (TVA) spectra obtained in acetonitrile for *E*-ES and *Z*-ES during the *E*→*Z* and *Z*→*E* photoisomerization following photoexcitation at their maximum absorption wavelength (321 and 318 nm, respectively). The data at the earliest time delay ($\Delta t$, see Methods for further detail) revealed a GSB feature appearing immediately following photoexcitation, which reflects the loss of the initial chromophore

ground-state ($S_0$) population in both *E*-ES and *Z*-ES starting isomers (the time evolution of this feature then reflects the recovery of the excited state population back to $S_0$). This GSB feature is centered at 1707 cm$^{-1}$ in *E*-ES and 1713 cm$^{-1}$ in *Z*-ES; comparison with the reported steady-state FTIR and density functional theory (DFT) calculations (discussed below) implies that these features are attributable to the bleach of the C=O stretching mode (see supplementary information Note S2 for further details).

In both isomers, a broad ESA stretching from the edge of the GSB to the low wavenumber region of the probe is also observed. These ESA band, which decay to baseline within pump-probe delay time ($\Delta t$) of 30 ps, could be associated with the IR-active mode of the vibrationally 'hot' $S_0$ molecules formed following IC from the excited-state. The presence of this feature from an early $\Delta t$ (i.e., near time-zero, $\Delta t = 0$) suggests that there may also be contributions from population of vibrational relaxation in the excited-state. To verify the origin of the ESA feature, we report in the supplementary information, Fig. S2, TVAS experiments analogous to the ones presented here using a weakly-interacting solvent, deuterated chloroform (CDCl$_3$). Previous literature has reported that interaction of solutes with a more strongly interacting solvent (polar, i.e., acetonitrile) will promote a higher rate of vibrational energy transfer in $S_0$ than in non-polar solvents (i.e., CDCl$_3$)[37,38]. As such, if the aforementioned ESA feature results from relaxation of vibrationally 'hot' $S_0$ molecule rather than vibrational relaxation in the electronically excited-state, the ESA feature should decay to baseline on a significantly longer timescale in CDCl$_3$ when compared to acetonitrile. In CDCl$_3$ the broad ESA feature decays to baseline on a similar $\Delta t$ of ~30 ps to that observed in acetonitrile. Together with the presence of the

ESA at near $\Delta t = 0$, this suggests that this feature is likely dominated by contributions from vibrational relaxation in the excited-state.

Finally, there is a second ESA band, centered at ~1717 cm$^{-1}$ and ~1708 cm$^{-1}$, for $E$-ES and $Z$-ES, respectively, which begins to grow at ~10 ps. Through comparison with the steady-state FTIR (see supplementary information Fig. S3), we assign these features to the photoinduced formation of either the $Z$ or $E$ isomer (for $E$ or $Z$ starting isomer, respectively). The deviation between the frequency of the C=O stretch in $Z$-ES in the steady-state and transient data (i.e., respectively, 1713 and 1717 cm$^{-1}$) could be attributed to the overlap between the starting $E$-isomer and the $Z$-photoisomer in the transient spectra. The experimental and computed difference IR spectra (see supplementary information Note S3 for explanation) reported in supplementary information Fig. S3 further support the aforementioned assignment. Furthermore, the spectral shifts in the GSB bands of the $E$-ES (red shifted) and $Z$-ES (blue shift) as the molecule relaxes back to the ground-state could be attributed to the overlap between the vibrational mode of the initial isomer and the photoisomer, in both cases. We also note that the stronger absorption intensity of the feature associated with $E$-ES photoisomer in Fig. 2b compared to the absorption of $Z$-ES photoisomer observed in Fig. 2a is due to (i) the stronger vibrational intensity of the $E$-ES isomer in the ground state as shown in Fig. 1c, and/or (ii) higher percentage recovery of the original $Z$-ES starting isomer (discussed later), such that convolution of the photoisomer (ESA) band with the GSB have minimal effect on the observed photoisomer intensity.

In addition to its ability to provide structural dynamics information in real-time, TVAS also allows for the estimation of the percentage recovery of the starting molecule (regardless of the pathway(s) facilitating that recovery). From the data reported in Fig. 2a, b), the percentage recovery for $E$ and $Z$ starting isomer is ~49% and ~94%, respectively. This suggests that, following photoexcitation, the relaxation pathway of $E$-ES produces about equal amount of $E$- and $Z$-isomer. Contrarily, the relaxation of $Z$-ES favors the formation of $Z$-isomer as the molecules return to the $S_0$.

However, as mentioned previously, in the case of $E$-ES and $Z$-ES studied herein, the photoisomer vibrational absorption bands might extend over the GSB and thus result in over-estimation of the bleach recovery. Hence, we have further evaluated the photoisomerization quantum yield (PQY) of both systems from the UV steady-state irradiation as reported in a previous study[39,40], and as shown in supplementary information Fig. S4 and discussed in Note S4. At the photostationary state (PSS) in acetonitrile, the calculated PQY for $E \rightarrow Z$ is 57% and $Z \rightarrow E$ is 23%. In both cases, we expect that the remaining % corresponds to the re-formation of the starting isomer. The disparity between the steady-state determined PQY of each starting isomer and the estimated percentage recovery for each starting isomer from the transient data (43% vs 49% for $E$-ES and 77% vs 94% for $Z$-ES, respectively) further confirms the spectral overlap in the TVAS measurement mentioned earlier. Altogether, these data reveal that, in acetonitrile, photoisomerization of $E$-ES is unbiased towards the formation of either isomer, as it produces about equal amount of $E$-ES and $Z$-ES as it returns to ground state. For $Z$-ES, photoisomerization is biased towards the re-formation of $Z$-ES. This is also evident in the changes observed in the UV absorption profile reported in supplementary information Fig. S4.

To extract the kinetic information from the TVA spectra of each isomer under study in acetonitrile, spectral decomposition of each dataset into individual absorption bands was carried out. This was achieved with three spectral basis functions using the KOALA software package (see supplementary information Fig. S5 and Note S5 for details)[41]. The kinetics of the three component

parts – GSB recovery, vibrational relaxation, and photoisomer formation– are shown in Fig. 2c, d, together with the mono-exponential fits and resulting time constants. There is no significant difference between the time constant extracted for the formation of the photoisomer in both $E$-ES and $Z$-ES, with both having values of $18.90 \pm 2.20$ ps and $18.72 \pm 1.80$ ps, respectively. These time constants compare well with those previously reported for photoisomerization in sinapate esters using transient electronic absorption spectroscopy (TEAS)[21,22,25,31,32,42]. Secondly, the time constants extracted for vibrational relaxation are $16.42 \pm 1.66$ ps for $E$-ES and $13.30 \pm 1.32$ ps for $Z$-ES. Finally, the time constants obtained for the GSB recoveries, which indicate repopulation of the starting isomer in their lowest vibrational level, are $9.05 \pm 0.84$ ps for $E$-ES and $7.33 \pm 0.66$ ps for $Z$-ES. These time constants are shorter than the time taken for the vibrational relaxation and formation of the photoisomers, and we discuss the possible cause and implications later on.

We corroborate the TVAS measurement with its electronic absorption equivalent technique (TEAS), and verified that the high sample concentration used in the present TVAS experiments has no influence on the dynamics observed with TEAS, as per comparison with previous work[24]. The resulting data is presented in supplementary information Fig. S6, and extracted time constants shown in supplementary information Table S3 compare well with those reported previously[21,24,31,32]. Also presented in supplementary information Fig. S7 and discussed in Note S7 is the steady-state fluorescence measurements for both isomers.

### Description of potential energies and conical intersections (CI).

Our experimental results naturally raise questions regarding the main factors controlling (i) the different PQY between the two starting isomers ($E \rightarrow Z = 57\%$; $Z \rightarrow E = 23\%$) and (ii) the much faster GSB recovery of the starting isomer (~7−9 ps) when compared to photoinduced isomerization (~18 ps). The topography of the potential energy surface (PES) and the CI shape are the main factors determining the split between different pathways in the excited-state[43,44]. Moreover, the probability of forming a photoproduct or regenerating the reactant is strongly correlated with the velocity and direction at which the molecule reaches the CI, which in turn determines its exit direction in the $S_0$ away from the intersection[45,46]. Therefore, to rationalize these intriguing experimental observations, we computed the excited-state PES and characterized the topography of the CIs using XMS-CASPT2 and CASSCF calculations in the gas phase. Computational details are reported in supplementary information Note S8 and Note S9 additional TDDFT results are given in the supplementary information Figs. S8–S10, and Tables S4–S6.

Optimizations of the $S_0$, $1^1\pi\pi^*$, and $2^1\pi\pi^*$ states of $Z$-ES and $E$-ES were carried out at CASSCF (6 electron in 6 orbitals) level, as shown in supplementary information Fig. S11. The energies of these structures were subsequently computed using a larger active space (12 electrons in 11 orbitals), as shown in supplementary information Fig. S12. Both small and large active space optimizations resulted in planar structures for both isomers (supplementary information Fig. S13 and Table S7). The electronic excitation to the bright $1^1\pi\pi^*$ (V state) state promotes one electron from a delocalized $\pi$−bonding orbital to a delocalized $\pi^*$−antibonding orbital. This excitation leads to a bond-length alternation (BLA) inversion (formal double bonds become single and vice-versa) weakening the C7–C8 bond (methine bridge). Although the BLA inversion promotes only an initial modest relaxation away from the Franck-Condon region, the acquired C7-C8 single-bond character enables the subsequent torsional motion towards the CI. The $2^1\pi\pi^*$ (V' state)

| **Table 1 Photoisomerization quantum yields and time constants in acetonitrile solution extracted from TVAS fitting.** | | | | | |
|---|---|---|---|---|---|
| Starting Isomer | $\Phi_E$ (%) | $\Phi_Z$ (%) | $\tau_{GSB}$ (ps) | $\tau_{vib.\ relaxation}$ (ps) | $\tau_{photoproduct}$ (ps) |
| $E$-ES | 43 | 57 | 9.1 ± 0.8 | 16.4 ± 1.7 | 18.9 ± 2.2 |
| $Z$-ES | 23 | 77 | 7.3 ± 0.7 | 13.3 ± 1.3 | 18.7 ± 1.8 |

has a different BLA pattern and lies energetically close to the $S_1$ state at XMS-CASPT2 level. However, it has much smaller oscillator strength (see supplementary information Table S8). The absolute and adiabatic energies are also presented in supplementary information Table S9.

The search for CIs associated with the photoisomerization was done with CASSCF(6,6) (see supplementary information Note S10). At this level, different structures, dihedral angles, and energies for the CI were located (supplementary information Fig. S14, Tables S10 and Table S11). The low energy $S_1/S_0$ CI structures mostly differ in the C6-C7-C8-C9 ($\phi$) and H-C7-C8-H (hydrogen out-of-plane, HOOP) dihedral angles, the key deactivation coordinates. The global minimum energy CI (MECI), termed $CI_{PYR}$ (or $CI_{PYR*}$, when $\phi$ rotates to the opposite direction), is nearly mid-way between the isomers (torsional angle ~80°) and presents pyramidalization in one of the methine atoms, akin to ethylene[47,48] and green fluorescent protein (GFP)[46]. Additionally, two other $S_1/S_0$ MECIs, close to the $E$- and $Z$-isomers, were found, termed $CI_E$ and $CI_Z$. Their energies are slightly above $CI_{PYR}$ (0.33 eV for $CI_E$ and 0.20 eV for $CI_Z$), both have a torsional angle nearest to their respective initial isomer compared to $CI_{PYR}$, but only $CI_E$ is pyramidalized at one of the methine atoms. These three intersections are likely connected through an intersection seam. Due to the respective geometric proximity of $CI_E$ and $CI_Z$ to the $E$ and $Z$ Franck-Condon regions, one may expect that IC occurs primarily at these intersections and not at $CI_{PYR}$.

The presence of energetic barriers connecting the $S_1$ minimum, and the CIs was investigated by interpolating these geometries using linear interpolation of internal coordinates (LIIC) as shown in supplementary information Figs. S15 and Fig S16. The $Z$ isomer's pathway to $CI_Z$ is barrierless, while a 0.1 eV barrier was found for the $E$ isomer's pathway to $CI_E$. Furthermore, the pathways connecting $CI_Z$ to $CI_{PYR}$ and $CI_E$ to $CI_{PYR}$ show barriers of 0.24 eV and 0.14 eV, respectively (see supplementary information Fig. S17). As such, no significant barrier for isomerization starting from $S_1$ of either isomer was found.

The topography of the PES in the vicinity of the $S_1/S_0$ lowest-energy CIs are discussed in supplementary information Note S12 and shown in Figs. S18 and S19. The crossing between the two surfaces (branching plane) is a function of the displacement along two coordinates: the gradient difference vector ($\vec{g}$) and the nonadiabatic coupling vector ($\vec{h}$)[44,49]. Thus, their relative orientation is indicative of the reactivity of the CI (see supplementary information Note S12 for details). In all CIs, $\vec{h}$ is mainly represented by a torsional motion of angle $\phi$, which promotes isomerization. The $+\vec{h}$ direction corresponds to a movement towards the starting isomer in $CI_E$ and $CI_Z$. The $\vec{g}$ vector in $CI_Z$ is dominated by skeletal deformation (mainly bond stretching along the BLA), while in $CI_{PYR}$ and $CI_E$ it is a combination of bond stretching and pyramidalization, governed by the fast HOOP mode induced upon torsion. Interestingly, the main contributions to $\vec{g}$ come from C=O, −C−CO, and C=C stretching modes, the same vibrations monitored in the TVAS measurements.

According to a classification proposed by Galvan et al.[50], the three CIs are sloped and single pathed (supplementary information Table S12) but the parameters describing these intersections are different. Note that $CI_E$ and $CI_Z$ are strongly sloped in the $\vec{g}$ direction (especially $CI_Z$) but peaked in the isomerization direction ($\vec{h}$), which would imply that no preferential isomer would be formed when leaving the CI in this direction. Furthermore, this classification is valid only in the immediate vicinity of the intersection point; the final outcome will be determined by the topography of the surface further away from the intersection. Our calculations indicate that one of the key points to explain the differences in the $Z$ and $E$ reactivity relies on the contribution of the HOOP mode to the exit direction of the CI, as discussed below[51].

## Discussion

By using TVAS, we have shown that the structural dynamics of sinapate esters leading to the formation of photoisomers can be directly observed in solution following UV excitation. When excited to the $1^1\pi\pi^*$ state, the relaxation back to the electronic ground-state is found to involve three major processes, namely, (i) repopulation of the starting isomer in the $S_0$, (ii) photoinduced isomerization, and (iii) vibrational relaxation. We will now discuss the implications of the extracted time constants for each process on the overall dynamics, drawing reference to the TEAS data (*viz* Table S3). The TVAS data are summarized in Table 1.

Starting with the GSB recovery for both molecules, which we have previously assigned to repopulation of the $S_0$, this reveals a time constant of 9.1 ± 0.8 ps for $E$-ES and 7.3 ± 0.7 ps for $Z$-ES meaning repopulation is very fast. This time constant reflects the dynamics attributed to repopulation of the lowest vibrational level of the electronic ground-state of the starting isomer. In contrast, the formation of the photoisomer in both $E$-ES and $Z$-ES occurs with a longer time constant, namely 18.90 ± 2.20 ps for $E$-ES and 18.72 ± 1.80 ps for $Z$-ES. These results suggest that smaller geometry rearrangements should be required to re-form the starting isomer than to form the photoisomer, which concurs with our calculations. Comparison between the decay time constants of previously reported TEAS measurements on sinapate esters and isomer-symmetric systems lends further support to this claim, with symmetric systems (systems for which there is symmetry around the $E/Z$ isomerization axis) rapidly decaying with comparable time constants to those reported for GSB recovery herein (i.e., ~10 ps)[4,33,34]. On the other hand, non-symmetric systems persist in the electronically excited-state for a longer period of time and decay with a similar time constant to the formation of photoisomer reported herein (~20 ps)[21,42]. To conclude our broad-stroke overview of our assignments for the extracted time constants, the time constant assigned to vibrational relaxation ($\tau_{vib.\ relaxation}$, see Table 1) is likely from trapped population in the electronically excited-state which traverses the CI, en route to photoisomer formation. This observation is also in keeping with our TEAS measurements, where a clear stimulated emission feature is present (see Fig. S6) which decays with a time constant comparable to the formation of the photoisomer.

There are two main aspects to be considered to explain the different reactivities of the isomers: (i) the starting isomer is always recovered faster than the photoisomer, and (ii) the PQY (%) at the PSS is 43:57 ($E$:$Z$) starting from $E$-ES, but it is 23:77 ($E$:$Z$) starting from the $Z$-ES, in acetonitrile.

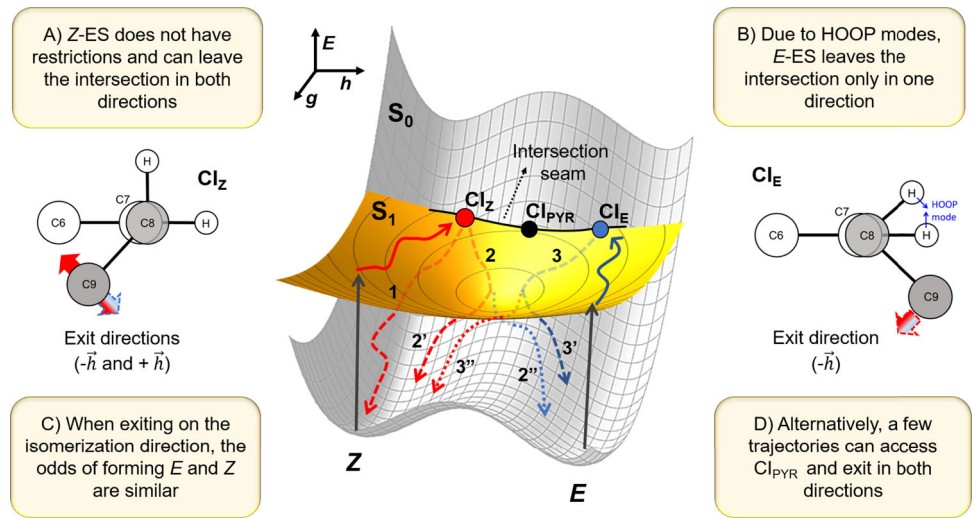

**Fig. 3 Topography of the relaxation pathways of *E*-ES and *Z*-ES.** Schematic representation of the CI geometries and potential energy surface (PES) for the photoisomerization starting from *Z* and *E* isomers. Solid red and blue lines indicate relaxation on the $S_1$ surface while dashed lines indicate relaxation in the ground state.

To uncover the origin of the CIs' different photoreactivity, we examined the PESs of the $S_1$ and $S_0$ states. Skeletal deformations along the BLA mode dominate the initial motion out of the Franck-Condon region. This relaxation unlocks the torsional motion, directing the population towards the crossing seam. Starting from *Z*-ES, the nearest $S_1/S_0$ crossing seam region is around $CI_Z$, while starting from *E*-ES, the nearest seam region is around $CI_E$ (see Fig. 3 for pictorial representation of $CI_{E/Z}$). Thus, one might expect these CIs to be the primary IC funnels (rather than $CI_{PYR}$, the lowest-energy CI). Recall that both $CI_Z$ and $CI_E$ have torsional angles closer to its starting isomer, which implies that smaller nuclear rearrangements are required to reform the initial isomer than to form the photoproduct. This proximity effect helps rationalize the first experimental puzzle: the fast recovery of the starting isomer compared to the photoisomer. If the IC happened through $CI_{PYR}$, we would expect similar times for both processes.

To explain the different PQY between *E*-ES and *Z*-ES starting isomer, we should examine the CIs' branching plane, defining the exit direction in $S_0$ immediately after IC. As we discussed, for all low-energy ES CIs, the branching plane is symmetric in the isomerization direction $\vec{h}$. Thus, when ES reaches the crossing seam, it is equally likely that it will exit in $S_0$ either toward $+\vec{h}$ or $-\vec{h}$. This split is what seems to happen when exciting *Z*-ES. After reaching $CI_Z$, it can exit in $S_0$ moving towards either the $Z$ $(+\vec{h})$ or $E$ $(-\vec{h})$ directions. This process is schematically shown in Fig. 3 as paths 1 and 2, respectively, coming out of $CI_Z$ in $S_0$. Due to the geometric proximity, the initial isomer is most probably recovered if it goes toward the $Z$ isomer (path 1). However, if it goes towards the $E$ isomer (path 2), the hot $S_0$ species can branch over the $E$-$Z$ barrier, either returning to $Z$ (path 2') or forming the photoproduct $E$ (path 2") with similar probabilities; the geometric proximity still causes $Z$ to be formed faster than $E$. Assuming that at $CI_Z$ the branching occurs with a nearly statistical distribution (50:50), then starting with *Z*-ES, the probability of recovering the starting isomer is ~75%, while the probability of forming the photoisomer is ~25%. We recall that the experimental results returned a PQY of 77% and 23% in acetonitrile, which is in excellent agreement with this computational estimate.

The situation is distinct when exciting *E*-ES, however. When this isomer reaches $CI_E$, the HOOP mode, which contributes to $\vec{g}$ direction, blocks the exit toward $E$ $(+\vec{h}$ direction in Fig. 3). As such,

*E*-ES can only exit in $S_0$ moving toward $Z$ (path 3). Similar to what happens with isomer $Z$, the hot $S_0$ species branches over the $E$-$Z$ barrier (paths 3' and 3"), yielding approximately 50:50 probability of obtaining the initial isomer or the photoisomer. Once again, this computational estimate is remarkably similar to the corresponding experimental results which, as discussed earlier, yielded a PQY of 43:57 for the $E:Z$ photoisomer in acetonitrile.

Indeed, the influence of HOOP vibrational modes in driving excited state population towards more reactive CIs has been investigated for other molecules capable of undergoing geometry photoisomerization[12,46,52]. When reaching the ground state at small velocities, a directional bias of the momentum along the faster HOOP mode can explain the different behaviors of excited sate population moving through CIs. Opposite directions of the HOOP mode would induce different outcomes before the slower rearrangement of the torsional modes occurs[46].

In summary, the different PQY are not explained by energetic barriers in the excited state but instead by the different reactivities of $CI_Z$ and $CI_E$. This remarkable finding agrees with the similar $\tau_1$ and $\tau_2$ extracted from the TEAS experiments shown in the supplementary information Note S6. While the $Z$-isomer can exit the intersection in two directions (one only reforming the $Z$ isomer and the other yielding both $Z$ and $E$), for *E*-ES, the HOOP mode directs the exit toward a single direction which yields $Z$ and $E$. Moreover, the torsional angle is closer to the original isomer at the exit of these CIs. Therefore, whichever the exit direction, the nuclear rearrangements required to reform the reactant's ground state are smaller than the ones required to form the photoisomer, explaining the faster GSB recovery of the starting isomer (compared to that of the formation of the photoisomer). Although this photoreactivity scenario is compelling and consistent with the experimental findings, a detailed description of the splitting of the decay channels and prediction of product branching ratios is only possible in nonadiabatic dynamics simulations, which are ongoing in our group.

We return to compare our TVAS data with the TEAS measurements reported in the supplementary information Fig. S6 for both *E*-ES and *Z*-ES in polar solvent (acetonitrile). The TEAS data revealed the presence of a stimulated emission feature which decays with a time constant of ~20 ps (supplementary information Table S3). Compared to the time constant reported for the GSB recovery in our TVAS measurement, this indicates that a fraction of population must remain in the excited-state for a

longer time. One possible explanation for this is the presence of a local minimum on the excited-state PES where population could be trapped. Although, our gas phase calculation revealed that there is no sizeable energy barrier on the $S_1$ PES towards the $CI_E$ or $CI_Z$, we note that this is likely to change in a solvated environment; the presence of the minimum on the PES may slow IC. Once the barrier is overcome, excited-state population proceeds towards (and through) the CI, resulting in isomerization on the ground-state.

Crucially, the large disparity between the time constant extracted for the GSB recovery and photoinduced isomerization in both isomers contradicts previous consensus on the photoisomerization process of sinapate esters[21,24,25,30–32]. The new results reported here reveal that the time constant previously assigned to photoisomerization (~20 ps), and previously thought to include both the formation of the starting isomer and the photoisomer, in fact only applies to the latter, since the starting isomer relaxes back to $S_0$ on a faster timescale.

As a caveat, while our calculations have been performed in the gas phase environment and agrees with the experimental data, we acknowledge that the inclusion of polar solvent in the calculation might present a potential energy surface and/or energy barrier that are slightly different from those reported herein. Moreover, we point out that the solvent would equally affect both isomers and with this, it would not play a role when it comes to explain the significant differences in the $E$ and $Z$ quantum yields (see supplementary information Table S2) or the different time scales of the ground state recovery and photoproduct formation, which are the main points in our discussion. Furthermore, the interpretation of the experimental data could further benefit from molecular dynamic calculation. However, to accurately describe this system, MRCI or XMS-CASPT dynamics would be required since CASSCF is unable to compute accurate energies and states order (see supplementary information Note S9). Those dynamical calculations are unaffordable for such large system and time scales reported on the experiments and are beyond the scope of the current work.

The present studies have shown the exquisite detail that time-resolved vibrational spectroscopy, along with computational calculations, can provide for elucidating the photoisomerization processes in small solvated organic molecules following UV excitation. Considering the extracted time constants for the individual excited/ground-state processes, we have been able to reveal that the relaxation mechanism of sinapate esters is a more complex process than currently reported; notably, we demonstrate that the formation of the starting isomer occurs on a strikingly faster time scale than photoinduced isomerization. Importantly, the results presented in this study could be used to explain the fast relaxation mechanism of symmetrically substituted sinapates and other molecules which not only explains the ultrafast dynamics of the molecules, but also the steady-state data. Crucially, the presented results provide potential insight into unravelling the dynamics of other systems such as photoswitches, molecular motors, photoreceptor proteins, phototherapy etc. where photoisomerization is thought to be the major deactivation mechanism.

## Methods
**Synthesis**. The $Z$-ES was synthesized using a Still–Gennari $Z$-selective olefination as previously reported[24]. $E$-ES was obtained in a two-step procedure: a sustainable proline-mediated Knoevenagel–Doebner condensation in ethanol leading to the sinapic acid from bio-sourced syringaldehyde[53], followed by a classical Fischer esterification in ethanol. Characterization of the samples were consistent with the literature[24].

**Steady-state measurements**. Separate samples of $E$-ES and $Z$-ES were prepared to a concentration of ~20 μM in acetonitrile, deuterated chloroform and cyclo-hexane. Ultraviolet/Visible (UV-Vis) spectra of these samples were taken in a 1 cm

path length quartz cuvette using a Cary 60 Spectrometer (Agilent Technologies), both before irradiation and at various times during single wavelength irradiation with an arc lamp (Fluorolog 3, Horiba). The samples were irradiated until PSS was reached. The irradiance at maximal absorption ($\lambda_{max}$) for each molecule was set to 'one sun' equivalent irradiance at Earth's surface, with an 8 nm full width half maximum (FWHM). FTIR spectra of each solution at a concentration of 20 mM were taken using an FTIR spectrometer (VERTEX 70 v, Bruker) before and after irradiation with solar simulator for 6 h. FTIR measurements were taken under a nitrogen environment to remove vibrational features associated with atmospheric gases. The same sample holder used for the TVAS measurements was employed to record the FTIR spectra over a wavenumber range of 500–4000 cm$^{-1}$ with a resolution of 1 cm$^{-1}$.

**Transient vibrational absorption spectroscopy (TVAS)**. The TVAS measurements were recorded at the Warwick Centre for Ultrafast Spectroscopy (go.warwick.ac.uk/WCUS). The experiment employed a UV pump and infrared (IR) probe beam and has been reported in detail previously[4,34,54]. We report only the information specific to the current experiment herein. TVAS measurements of $E$-ES and $Z$-ES were taken in acetonitrile and deuterated chloroform at 20 mM concentration. In all cases, the pump excitation wavelength was chosen to match the relevant $\lambda_{max}$ with a mid-IR probe pulse centered at ~1700 cm$^{-1}$. The sample was delivered through a demountable Harrick Scientific flow-through cell equipped with two CaF$_2$ windows separated by 150 μm polytetrafluoroethylene (PTFE) spacers, thereby defining the optical path length of the sample. The samples were circulated using a diaphragm pump (SIMDOS, KNF) recirculating from a 25 mL sample reservoir to ensure each pump-probe pulse interacts with fresh sample, with a maximum pump-probe time delay ($\Delta t$) of 1 ns. The Harrick Scientific flow-through cell is mounted on a translation stage and translated throughout the data collection to prevent formation of products on the CaF$_2$ window.

**Transient electronic absorption spectroscopy (TEAS)**. The capabilities of the femtosecond (fs) TEAS setup and procedure employed in this study have been reported in detail previously[4,23,33,55,56], and only information specific to the present experiment is reported here. The same sample concentration used in the TVAS measurements (~20 mM) was employed for the TEAS measurements. In all cases, the pump excitation wavelength was chosen to match the relevant $\lambda_{max}$; the probe consisted of a broadband white light continuum pulse, with a maximum pump-probe time delay of 2 ns. The same sample delivery system as in the TVAS measurements was employed, but with a 6 μm pathlength.

*Computational methods*
Frequency calculations: Density functional theory (DFT) geometry optimization was performed at B3LYP/6-311++G** level of theory on the structures of $E$-ES and $Z$-ES, using NWChem software[57]. Following geometry optimization, the most stable conformer of each isomer was selected for frequency calculation in the ground-state at B3LYP/6-311++G**. In all calculations, the conductor-like screening model (COSMO) was used to model the effect of solvent dielectric parameters[58,59]. The default COSMO solvent model for acetonitrile within NWChem was used, descriptors of which are based on the Minnesota Solvent Descriptor Database[60].

Exploration of the PES and conical intersections: The potential energy surfaces of $Z$-ES and $E$-ES were investigated using multiconfigurational methods. Detailed information is given in the supplementary information Note S8–S12. Briefly, the geometries of the $S_0$, $S_1$ and $S_2$ states were optimized using SA4-CASSCF(6,6)/6-31Gd), and the geometries of the CIs were optimized using SA2-CASSCF(6,6)/6-31G(d). Single point calculations at these geometries were carried out using XMS-CASPT2(12,11)/6-31G(d). Linear interpolated geometries in the internal coordinates were performed at the same level. All multiconfigurational calculations used OpenMolcas v.19.11 (tag 283-ge7efbbb)[61].

## Data availability
The datasets presented in this study can be found in online repositories. The names of the repository/repositories and accession number(s) can be found below: Zenodo repository https://doi.org/10.5281/zenodo.7126478.

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

## Acknowledgements

The authors acknowledge the FET-Open grant BoostCrop (Grant agreement 828753) and the Warwick Centre for Ultrafast Spectroscopy (WCUS) for the use of transient vibrational absorption spectroscopy (TVAS) and Fluorolog 3 apparatus. T.T.A. thanks the University of Warwick for a Ph.D. studentship through the Chancellor Scholarship. J.M.T., M.T.do.C. and M.B. acknowledge Centre de Calcul Intensif d'Aix-Marseille for granting access to its high-performance computing resources. This work was granted access to the HPC resources of TGCC under the allocation 2021-A0110813035 made by GENCI. B.R., A.L.F and F.A. thank the Agence Nationale de la Recherche (grant number ANR-17-CE07-0046), as well as the Grand Reims, Conseil Départemental de la Marne, and the Grand Est region for financial support. V.G.S. thanks the Royal Society for a Royal Society Industry Fellowship. The authors thank Dr Michael Grubb for guidance on the use of KOALA, suggestions and for proof reading. The authors thank Dr. Natércia d. N. Rodrigues and Ana González Moreno for proof reading.

## Author contributions

T.T.A. acquired and analyzed the time-resolved and steady-state spectroscopic data and prepared the first draft of the manuscript. D.M. and J.M.W. contributed to the time-resolved spectroscopic data analysis. T.T.A., D.M. and J.M.W. interpreted the time-resolved spectroscopic data. T.T.A. computed and interpreted the vibrational frequencies of the molecules for assignment of steady-state vibrational spectral. J.M.T., M.Tdo C. and M.B. conducted and interpreted the computational data. A.F., B.R. and F.A. performed the synthesis and characterization of the samples. V.G.S. conceived the experiment and guided the accompanying data analysis and interpretation. All authors contributed to writing of the manuscript.

## Competing interests

The authors declare no competing interests.
