## [Peer Review File · Communications Chemistry]

This manuscript has been previously reviewed at another Nature Portfolio journal. This document only contains reviewer comments and rebuttal letters for versions considered at Communications Chemistry.

REVIEWERS' COMMENTS:

Reviewer #1 (Remarks to the Author):

The authors have addressed all issues raised in the previous round of review and the manuscript warrants publication in Communications Chemistry as-is.

With respect to the issue of whether isomerization occurs on the ground or excited state: The results and analysis contained in the current study may not rise to the level of 'definitive' regarding this issue, but rather, are a useful contribution to the literature of these molecular systems. The most important issue for this particular manuscript is that the interpretation of the experimental results (and computations) are consistent and reasonable given the the observations/simulations. Even if this interpretation ends up being incorrect, that would be for subsequent work to determine.

That said, the authors would do well to acknowledge this lack of certainty in some of their wording. For example:

L23: "Our mechanistic analysis reveals that, "

should be: "Our mechanistic analysis predicts that, "

and

L25: "We demonstrate that the isomerization yield is strongly dependent on the different topographies of CI near each of the initial (E/Z) isomer geometry"

should be:

"Our calculations suggest that different CI topographies at relevant points on the seam of intersection may influence the isomerization yield"

This more careful wording is recommended, but ultimately left to the authors as optional revision.

Reviewer 1

We would like to thank **Reviewer 1** for their comments. Our responses are documented below with our actions in the revised manuscript highlighted in red.

The authors have addressed all issues raised in the previous round of review and the manuscript warrants publication in Communications Chemistry as-is.

With respect to the issue of whether isomerization occurs on the ground or excited state: The results and analysis contained in the current study may not rise to the level of 'definitive' regarding this issue, but rather, are a useful contribution to the literature of these molecular systems. The most important issue for this particular manuscript is that the interpretation of the experimental results (and computations) are consistent and reasonable given the observations/simulations. Even if this interpretation ends up being incorrect, that would be for subsequent work to determine.

That said, the authors would do well to acknowledge this lack of certainty in some of their wording. For example:

L23: "Our mechanistic analysis reveals that, "

should be: "Our mechanistic analysis predicts that, "

and

L25: "We demonstrate that the isomerization yield is strongly dependent on the different topographies of CI near each of the initial (E/Z) isomer geometry"

should be: "Our calculations suggest that different CI topographies at relevant points on the seam of intersection may influence the isomerization yield"

This more careful wording is recommended, but ultimately left to the authors as optional revision.

Response: We agree that careful wording recommended by the Reviewer is needed to highlight the lack of certainty. We have now made changes to the manuscript as shown below.

Action: Manuscript page 1, Line 23 now reads "...Our mechanistic analysis **predicts** that, from the photoprepared state..."

Manuscript page 1, Line 25 now reads "...of the initial isomer. **Our calculations suggest that different CI topographies at relevant points on the seam of intersection may influence the isomerization yield.**"

We would once again like to thank **Reviewer 1** for their valuable feedback.